# Synthesis of Si-Based High-Efficiency and High-Durability Superhydrophilic-Underwater Superoleophobic Membrane of Oil–Water Separation

**DOI:** 10.3390/ma14102628

**Published:** 2021-05-18

**Authors:** Xiao-Hui Fang, Su-Hui Chen, Lan-Lin Yi, Zhong-Bin Yin, Yong-Jun Chen, Hong Jiang, Chang-Jiu Li

**Affiliations:** 1State Key Laboratory of Marine Resource Utilization in South China Sea, Special Glass Key Lab of Hainan Province, Hainan University, Haikou 570228, China; fangxiaohui110@sina.com (X.-H.F.); csuhui15@gmail.com (S.-H.C.); yilanlin20@gmail.com (L.-L.Y.); yinzhongbin110@sina.com (Z.-B.Y.); yongchen@hainanu.edu.cn (Y.-J.C.); jhong63908889@sina.com (H.J.); 2Key Laboratory of Advanced Materials of Tropical Island Resources of Ministry of Education, Haikou 570228, China

**Keywords:** one-step hydrothermal process, SiO_2_-Na_2_SiO_3_-coated SSF, oil–water separation, excellent performance

## Abstract

Oil pollution is caused by the frequent discharge of contaminated industrial wastewater and accidental oil spills and is a severe environmental and health concern. Therefore, efficient materials and processes for effective oil–water separation are being developed. Herein, SiO_2_-Na_2_SiO_3_-coated stainless steel fibers (SSF) with underwater superoleophobic and low-adhesion properties were successfully prepared via a one-step hydrothermal process. The modified surfaces were characterized with scanning electron microscopy (SEM), and contact angle measurements to observe the surface morphology, confirm the successful incorporation of SiO_2_, and evaluate the wettability, as well as with X-ray diffraction (XRD). The results revealed that SiO_2_ nanoparticles were successfully grown on the stainless-steel fiber surface through the facile hydrothermal synthesis, and the formation of sodium silicate was detected with XRD. The SiO_2_-Na_2_SiO_3_-coated SSF surface exhibited superior underwater superoleophobic properties (153–162°), super-hydrophilicity and high separation efficiency for dichloromethane–water, n-hexane–water, tetrachloromethane–water, paroline–water, and hexadecane–water mixtures. In addition, the as-prepared SiO_2_-Na_2_SiO_3_-coated SSF demonstrated superior wear resistance, long-term stability, and re-usability. We suggest that the improved durability may be due to the presence of sodium silicate that enhanced the membrane strength. The SiO_2_-Na_2_SiO_3_-coated SSF also exhibited desirable corrosion resistance in salty and acidic environments; however, further optimization is needed for their use in basic media. The current study presents a novel approach to fabricate high-performance oil–water separation membranes.

## 1. Introduction

The massive discharge of industrial wastewater contaminated with oil, oil pollution due to daily human activities, and accidental oil spills in water systems are major environmental and health concerns [1,2,3,4,5,6,7,8,9]. Thus, there is an urgent need for novel materials and processes to treat oil-containing wastewater. Accordingly, several oil–water separation techniques based on adsorption, sand filtration, evaporation, oxidation, electrochemical, photocatalytic treatment, and room-temperature ionic liquid treatment, have been developed by different research groups [10,11,12,13,14]. However, most of these traditional methods incur high costs, employ toxic compounds, require large areas for installation and processing, and generate secondary pollutants. Therefore, it is of utmost significance to design functional materials for low-cost and high-efficiency oil–water separation processes [15,16,17].

Recently, membrane technologies have garnered significant research attention for efficient oil–water separation [18]. Membrane separation processes have several advantages over traditional oil–water separation methods including simplicity, low energy consumption, and high efficiency [19,20,21,22]. Oil–water separation membranes are usually superhydrophobic–superoleophilic membranes; however, such lipophilic membrane materials are easily blocked by oil during the separation process, which reduces the effective oil flux through the membrane. By comparison, underwater superoleophobic membranes do not suffer from this same limitation, and research on developing novel underwater superoleophobic membranes is of great significance [23,24,25]. However, the successful realization of superoleophobic membranes for oil–water separation is hindered by different challenges. For example, Li et al. created a layered-structure by chemical etching a stainless-steel mesh and coating it with PVA-Na_2_SiO_3_. The resulting material was a hydrophilic mesh that reached 97% oil–water separation efficiency after 10 cycles of oil–water separation [26]. Gou et al. reported a simple method based on immersing a stainless steel substrate into a cement solution to create an SSM; however, the oil–water separation efficiency decreased to only 95% after five cycles [27]. Zhou used dip coating to deposit hydrophobic VPOSS (Octavinyl-polyhedral oligomeric silsesquioxane) nanoparticles onto a surface and then grafted the surface with a hydrophilic 2-mercaptoethyl containing carboxyl group (containing carboxyl group) to prepared a CF membrane. The membrane can be used for oil–water separation. After 10, 20, 30, 40 and 50 separation cycles, the separation efficiency reached 99.1%, 98.4%, 97.6%, 96.5% and 95.2% [28], respectively. Wang et al. grafted Kaolin particles onto a stainless-steel mesh to form a filter with a unique, micro-rough surface with connected pores, thus providing a path for quick gravity-driven oil–water separation. However, after 10 oil–water separation cycles, the separation efficiency dropped to only 97.2% [29]. The unique surface structure was destroyed by liquid flowing through the membrane due to an adverse interaction between the nanoparticles and the matrix, resulting in a decrease in the oil–water separation efficiency over use. Moreover, the nanoparticles had a smaller size scale and were not biodegradable [30,31,32]. Therefore, there is a great need for stable and environmentally friendly superhydrophilic and underwater superoleophobic materials for effective, long-lasting, and environmentally friendly oil/water separation.

Wang et al. deposited PDA (polydopamine) particles in situ to generate surface roughness, and then enhanced the surface hydrophilicity and underwater lipophilicity of the membrane by adsorbing hydrophilic chitosan to the surface [32]. Yang et al. coated the surface of a cellulose membrane with sodium alginate and then assembled CaCO_3_ particles on the surface using ASP (alternating soaking process) to obtain a superhydrophilic/underwater superoleophobic cellulose membrane for oil/water separation [30]. You et al. prepared a phosphorylated polyvinyl alcohol (PPVA) hydrogel and then sprayed the hydrogel with an aqueous metal ion to quickly crosslink the PPVA, producing a superhydrophilic material with underwater superoleophobicity [33]. Qing et al. prepared hydrophilic PVA (Polyvinyl alcohol) nanofiber membranes by electrospinning and then grew silica nanoparticles on the nanofibers using an in situ, modified Stöber reaction [34]. Wang et al. modified a PVDF-HFP (polyvinylidene fluoride-co-hexafluoropropyle) tubular nanofiber membrane with Ag nanoparticles by co-depositing CA (catechol) and PEI (polyethyleneimine) on the surface and then adding KH560 (glycidyloxy propyltrimethoxysilane) to obtain a membrane with super-hydrophilicity and good antifouling properties [35].

Herein, SiO_2_ nanocrystal coated stainless steel fibers (SSF) are fabricated using a facile, one-step hydrothermal process to construct a micro-nano surface with special wetting properties. The SiO_2_-Na_2_SiO_3_-coated SSF showed excellent superhydrophilicity in air and superoleophobicity underwater for different oils. The as-prepared SiO_2_-Na_2_SiO_3_-coated SSF showed excellent performance in oil–water separation tests, sandpaper abrasion tests, long-term stability tests, and after separation cycle tests. Moreover, the water flux through the membranes reached 76,433 L m^−2^ h^−1^. The current study shows that the newly created SiO_2_-Na_2_SiO_3_-coated SSF are promising candidates for oil–water separation.

## 2. Experimental Section

### 2.1. Materials and Synthesis

Tetraethyl silicate, ethanol, NaOH, n-hexane, HCl, and NaCl (analytical grade) were purchased from Shanghai Maclin Biochemical Technology Co., Ltd., Shanghai, China. Kerosene, hydraulic oil (“Great Wall” Brand, Beijing, China), machine oil (“Great Wall” Brand, Beijing, China), soybean oil (“Arawan” Brand), and SSF (Fe, Al, Ni) were purchased from the local supermarket. The diesel was obtained from the local oil station.

Briefly, 75 mL of ethanol, 25 mL of water, 4.5 mL of tetraethyl silicate and 12.15 g of sodium hydroxide were added into a beaker and magnetically stirred for 20 min, followed by pouring into the reaction kettle. The stainless-steel mesh was cut into 4 cm × 4 cm pieces, cleaned with ethanol and placed in a reaction kettle with the above solution. Then, the reaction kettle was placed in a hydrothermal oven at 95 °C for 40 min. After 40 min, SSF were taken out, and the SSF were washed with water and dried in an oven at 85 °C for 24 h.

### 2.2. Characterization

The surface morphology of SiO_2_-Na_2_SiO_3_-coated SSF was observed by a field emission scanning electron microscope (FESEM; TESCAN MIRA3, Brno, Czech Republic). The contact angle was measured by using an optical contact angle meter (Drop meter A-100, Maishi Testing Technology Co., Ltd., Ningbo, China) at ambient temperature. The volume of oil droplet was ~5 µL, X-ray diffraction (XRD, Bruker D8 Advance X, Aachen, Germany) analysis was performed on a Bruker D2 X-ray diffractometer, equipped with Cu Kα radiations (40 kV, 40 mA).

### 2.3. Oil/Water Separation

SiO_2_-Na_2_SiO_3_-coated SSF was fixed between two glass devices, which were pre-soaked with water. The oil/water mixture (v:v = 1:1) was slowly poured into the glass tube and permeated liquid was collected by using a beaker. Herein, we utilized engine oil, hydraulic oil, soybean oil, n-hexane, diesel oil and kerosene. In the separation process, the water rapidly passed through the treated fabric, whereas the oil was repelled by the fabric. The separation efficiency (E) was calculated by using the given relationship [27,32,36]:E = (m_1_/m_0_ × 100%)(1)
where m_0_ and m_1_ respectively represent the weight of water in the original mixture and the weight of water collected after separation. The permeation flux of as-prepared SiO_2_-Na_2_SiO_3_-coated SSF was calculated by using the permeate volume per unit time, as given below [27,32,37]:Flux = V/(A × t)(2)
where V represents the permeate volume (L), A denotes the effective filtration area of the as-prepared Ti-foam (m_2_) and t refers to the filtration time (h).

### 2.4. Corrosion Resistance

The corrosion resistance of SiO_2_-Na_2_SiO_3_-coated SSF was measured by using oil-corrosive aqueous solution separation tests of hexane-3.5 wt % NaCl, saturated NaCl, 1 M HCl and 1 M NaOH [38].

### 2.5. Sandpaper Abrasion Test

A sandpaper abrasion test was carried out to investigate the mechanical stability of SiO_2_-Na_2_SiO_3_-coated SSF. The SiO_2_-Na_2_SiO_3_-coated SSF (4 cm × 4 cm) was placed against sandpaper (grit No. 1000) under a 100 g weight and an external force was applied to move 10 cm away from the initial position and return back. This process was repeated 100 times [39].

### 2.6. Long-Term Stability

To assess the long-term stability, SiO_2_-Na_2_SiO_3_-coated SSF was immersed in water, maintained at 60 °C, for two weeks. The stability was assessed by evaluating the underwater oil contact angle and oil antifouling (self-cleaning) performance [40].

## 3. Results and Discussion

### 3.1. SEM Analysis

Figure 1a–d presents the SEM images of the as-received and SiO_2_-Na_2_SiO_3_-coated SSF. Figure 1a,b shows that the as-received SSF had a smooth surface, and the fibers were hydrophobic with a water contact angle of ~146.5°. As expected, the SEM image of SiO_2_-Na_2_SiO_3_-coated SSF revealed the presence of a large number of SiO_2_ particles Figure 1c,d that were uniformly distributed on the surface, creating a microscopically rough structure. Figure 1d shows a high-magnification SEM image of the SiO_2_-Na_2_SiO_3_-coated SSF with a hierarchical microstructure formed by a large number of SiO_2_ nanoparticles. Moreover, this SiO_2_ coating influenced the wettability of the matrix surface, as shown in the insets of Figure 1c,d. The SiO_2_-Na_2_SiO_3_-coated SSF were super hydrophilic (~0°) and superoleophobic (~160.5°) in air and underwater, respectively. Herein, dichloromethane was utilized as a model oil.

### 3.2. EDS and XRD Analysis

The surface elemental composition of the oil–water separation membrane was studied with EDS. As shown in Figure 2a, in addition to the metal elements from the stainless steel fibers, silicon, oxygen, and sodium, were also detected on the surface, which supported that the SiO_2_ and Na_2_SiO_3_ were successfully deposited. X-ray powder diffraction was used to further confirm that the fiber surface was coated with SiO_2_-Na_2_SiO_3_. Figure 2b shows the XRD pattern of the oil–water separation membrane measured by the X-ray powder diffraction method. The blue curve represents the data for the prepared oil–water separation membrane, and the green and red curves represent the standard cards for SiO_2_ and Na_2_SiO_3_, respectively. From the data, we see that the characteristic peaks of SiO_2_ appeared at 21.03°, 26.7°, 36.7°, 39.55° and other positions, which supported that the SiO_2_ was formed in the mixed powder. The characteristic peaks of Na_2_SiO_3_ appeared at 21.03°, 28.87°, 30.66°, 33.85° and other positions, supporting that Na_2_SiO_3_ was also present in the sample. From the EDS and XRD analysis, we can conclude that Si^4+^ was hydrolyzed to produce SiO_2_, the SiO_2_ reacted with sodium hydroxide to produce Na_2_SiO_3_, and SiO_2_ and Na_2_SiO_3_ self-assembled to obtain SiO_2_-Na_2_SiO_3_ coated oil–water separation membrane during the synthesis procedure.

### 3.3. Surface Wettability

Figure 3a,b presents the contact angle (CA) and sliding angle (SA) of underwater oil droplets on the surface of SiO_2_-Na_2_SiO_3_-coated SSF membranes. We selected dichloromethane, hexadecane, n-hexane, paraffin oil, and carbon tetrachloride as model oils to prepare different oil–water mixtures for measuring the contact angle of underwater oil. Selected dichloromethane, carbon tetrachloride, Chloroform, C6F15N and dichloroethylene were used as model oils to measure the rolling angle of underwater oil. The SiO_2_-Na_2_SiO_3_-coated SSF membranes were immersed in water, and the oil wettability was characterized by measuring the underwater oil CA and SA. The SiO_2_-Na_2_SiO_3_-coated SSF membranes exhibited underwater superoleophobicity to the different oils with an oil CA of >150 and SA < 10°. The wettability of the original and SiO_2_-Na_2_SiO_3_-coated SSF was evaluated by the water and oil contact angle (CA) and sliding angle (SA) measurements. As shown in Figure 3c, when a water droplet (5 µL) touched the original SSF, a hydrophobic behavior was observed. On the other hand, when a water droplet (5 µL) was dropped onto the SiO_2_-Na_2_SiO_3_-coated SSF membrane, it quickly spread with a CA of almost 0°, as shown in Figure 3d. Therefore, the SSF surface changed from hydrophobic to superhydrophilic after the surface treatment and had underwater superoleophobic properties.

### 3.4. Oil and Water Separation Efficiency Test

When separating oil and water, it was observed that the water droplets permeated through SiO_2_-Na_2_SiO_3_-coated SSF membrane and reached the glass container under the action of gravity, whereas the oil was retained on top of the membrane due to the fact that underwater super oleophobicity of the membrane makes the oil impenetrable, as shown in Figure 4d. We use the above Equation (1) to calculate the oil–water separation efficiency. The separation efficiency of various oil/water mixtures were shown in Figure 4a. For all selected oils, the separation efficiency is greater than 99%. As shown in Figure 4b, after 50 cycles, the separation efficiency of the SiO_2_-Na_2_SiO_3_-coated stainless steel fiber still maintained very high (>99%), showing good oil–water separation performance and, as shown in Figure 4c, the SiO_2_ nanoparticles are still present on the surface.

### 3.5. The Water Flux Test

The Equation (2) was used to calculate the water flux. Herein, a high water flux of 76,433 L m^−^^2^ h^−^^1^ is rendered by SiO_2_-Na_2_SiO_3_-coated SSF. We also conducted 50 oil–water separations and observed the changes in water flux with the number of separations. As shown in Figure 5a. after 50 oil–water separations, there is still a large water flux. One should note that self-cleaning is another essential indicator of coating stability. As shown in Figure 5b, the original SSF is stained with oil droplets, and the oil droplets do not float when the original SSF is put into water, whereas the oil droplets are quickly separated in the case of SiO_2_-Na_2_SiO_3_-coated SSF, therefore, the SiO_2_-Na_2_SiO_3_-coated SSF has self-cleaning property in water [41].

### 3.6. The Abrasion Test

Furthermore, the mechanical stability of SiO_2_-Na_2_SiO_3_-coated SSF was investigated by evaluating the separation capability after the abrasion test, as described in the experimental section. As shown in Figure 6a, the separation efficiency of SiO_2_-Na_2_SiO_3_-coated SSF remained higher than 99% for different oil/water mixtures. Compared with Figure 1c,d, Figure 6b shows that the density of SiO_2_ nanoparticles was significantly reduced due to the abrasion, however, still, a reasonable number of SiO_2_ particles remained attached to the surface of SiO_2_-Na_2_SiO_3_-coated SSF.

### 3.7. Long-Term Stability Test

The stability of the SiO_2_ coating layer plays a vital role in the long-term stability of the separation membrane. Therefore, we evaluated the separation efficiency of SiO_2_-Na_2_SiO_3_-coated SSF, immersed in hot water for 2 weeks, to assess the durability of as-prepared SiO_2_-Na_2_SiO_3_-coated SSF. Figure 7a shows the separation efficiency of SiO_2_-Na_2_SiO_3_-coated stainless steel fiber for different oil/water mixtures under long-term immersion in hot water at 60 °C. It can be readily observed that the separation efficiency remained higher than 99%, indicating the stable performance of as-prepared SiO_2_-Na_2_SiO_3_-coated SSF in an aqueous environment. Moreover, the morphology and distribution of SiO_2_ nanocrystals did not change after being immersed in hot water, as shown in Figure 7b.

### 3.8. Corrosion Resistance Test

Lastly, we examined the corrosion resistance of as-prepared SiO_2_-Na_2_SiO_3_-coated SSF by evaluating the separation efficiency of an oil-corrosive aqueous solution, which has been prepared by independently mixing hexane (dyed in red with red O) with 1 M NaCl, 1 M HCl and 1 M NaOH. The results reveal that hexane-1 M NaCl and hexane-1 M HCl mixtures were completely separated, as shown in Figure 8a,b, after 50 oil–water separations, the separation efficiency is still greater than 99%, indicating excellent resistance towards salt and acid environments. However, in the case of hexane–NaOH solution, the color of the filtrate turned blue, which implies that the as-prepared SiO_2_ nanocrystals were damaged in the presence of a base. a small amount of oil (hexane) passed through SiO_2_-Na_2_SiO_3_-coated SSF and permeated the filtrate, which indicates that the base resistance of SiO_2_-Na_2_SiO_3_-coated SSF should be further improved, which may be caused by the existence of a chemical reaction between SiO_2_ and NaOH.

### 3.9. Principle of Strong Membrane Bonding

Figure 9a illustrates the steps in an ordinary oil–water separation membrane separation process. In a typical membrane, the surface structure is destroyed after the separation, which causes the separation efficiency to drop. Figure 9b illustrates the separation using a superhydrophilic-underwater superoleophobic membrane prepared by a one-step hydrothermal method. The precursor ethyl orthosilicate was cracked to form Si^4+^ under alkaline conditions, and SiO_2_ was formed by a hydrolysis reaction [42]. The SiO_2_ continued to react with NaOH to form sodium silicate. However, the reaction between SiO_2_ and NaOH was slow, thus a mixed solution of SiO_2_ and sodium silicate could be prepared by controlling the reaction time. After drying, the SSF coated with a SiO_2_-Na_2_SiO_3_ film were formed. Sodium silicate is a good adhesive with excellent film-forming properties. It is worth mentioning that SiO_2_ is a low-cost, environmentally friendly material, which is very suitable for a wide range of applications. Moreover, after 50 cycles, the oil–water separation efficiency was still above 99%. The reason for the good performance was that the sodium silicate was not only adhered physically to the fibers, but the SiO_2_ and Fe_2_O_3_ were adsorbed onto the surface through chemical, electrostatic, and hydrogen bonding interactions. Therefore, the SiO_2_ was strongly bound to the SSF [43,44,45].

## 4. Conclusions

In summary, a SiO_2_-Na_2_SiO_3_-coated stainless steel fiber (SSF) membrane with superior underwater superoleophobic behavior was successfully fabricated using a simple, one-step hydrothermal route. The original and SiO_2_-Na_2_SiO_3_-coated SSF membranes had different surface topographies and roughness, resulting in superior oil–water separation performance in the coated membrane. The SiO_2_-Na_2_SiO_3_-coated SSF exhibited a high separation efficiency of dichloromethane–water, n-hexane–water, tetrachloromethane–water, paroline–water, and hexadecane–water mixtures. Moreover, the SiO_2_ layer remained stable after being immersed in hot water (60 °C) for two weeks and demonstrated excellent wear resistance. In addition, the SiO_2_-Na_2_SiO_3_-coated SSF maintained a high separation efficiency of >99% after 50 separation cycles. Lastly, the SiO_2_-Na_2_SiO_3_-coated SSF membranes exhibited superior salt and acid resistance; however, the as-prepared separation membrane was not stable in a basic medium. The successful realization of SiO_2_-Na_2_SiO_3_-coated SSF membranes affords a wide range of potential applications, including underwater oil transport, flexible and wearable oil-repellent, and oil/water separation materials.

## Figures and Tables

**Figure 1 materials-14-02628-f001:**
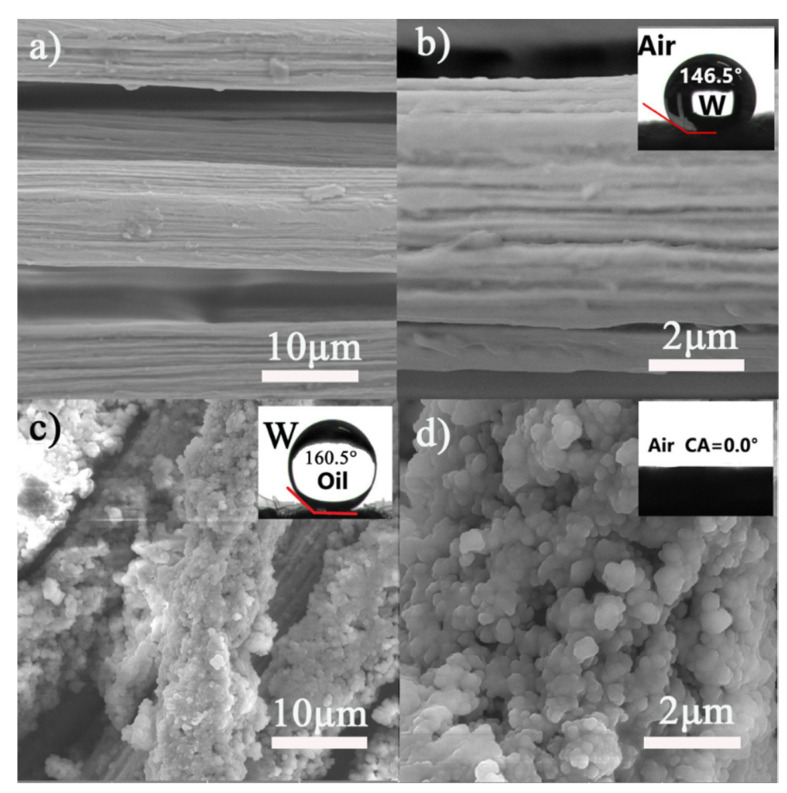
SEM images of (**a**,**b**) original stainless steel fibers (SSF) and SiO_2_-Na_2_SiO_3_-coated (**c**,**d**) SSF. The insets show the contact angle (**b**) of raw SSF, treated underwater oil contact angle (**c**) and water contact angle (**d**) in air.

**Figure 2 materials-14-02628-f002:**
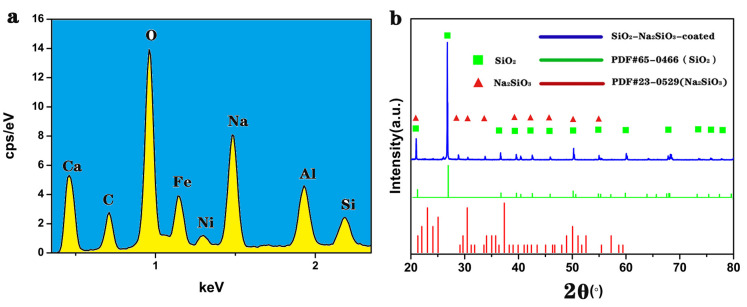
(**a**) EDS spectrum of the SiO_2_-Na_2_SiO_3_-coated SSF and (**b**) the XRD diffraction patterns.

**Figure 3 materials-14-02628-f003:**
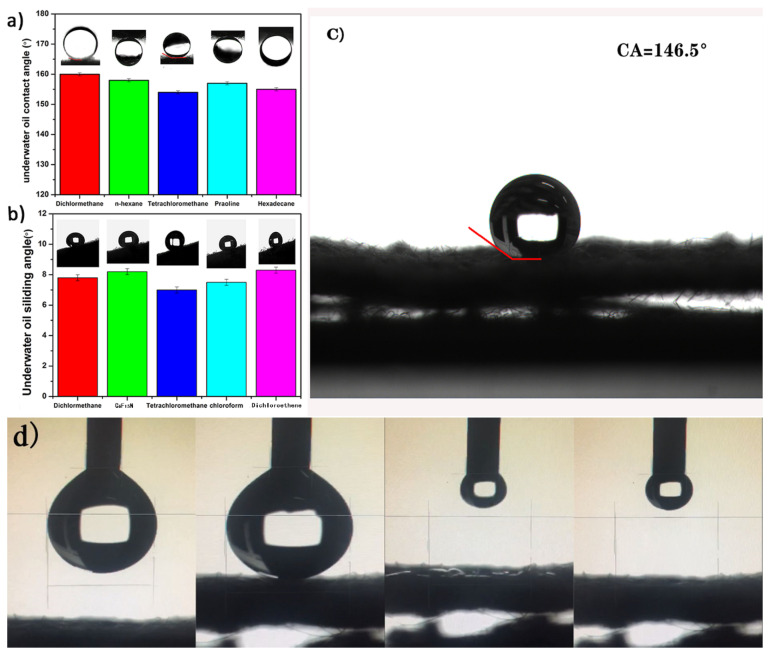
The underwater contact (**a**) angle and sliding angle (**b**) of different oil droplets on the SiO_2_-Na_2_SiO_3_-coated SSF membrane, the hydrophobicity (**c**) of original SSF and the superhydrophilicity (**d**) of SiO_2_-Na_2_SiO_3_-coated.

**Figure 4 materials-14-02628-f004:**
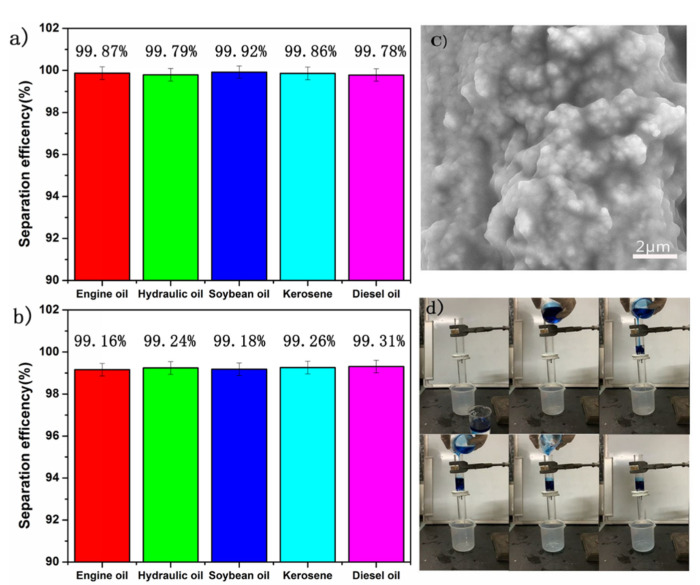
The separation efficiency of different oil/water mixtures (**a**) before and (**b**) 50 separation cycles SEM image (**c**) of SiO_2_-Na_2_SiO_3_-coated SSF after 50 separation cycles with the separation process (**d**).

**Figure 5 materials-14-02628-f005:**
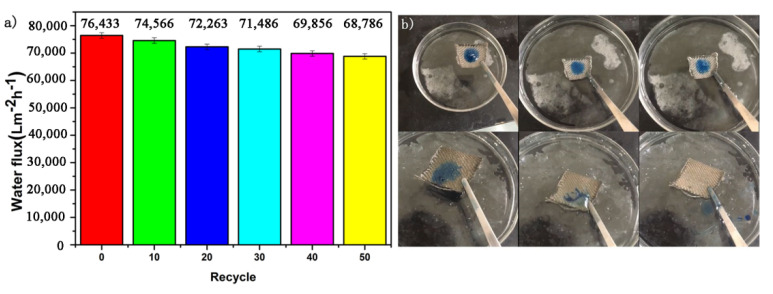
Variation of water flux with the number of recycles (**a**) and self-cleaning performance (**b**).

**Figure 6 materials-14-02628-f006:**
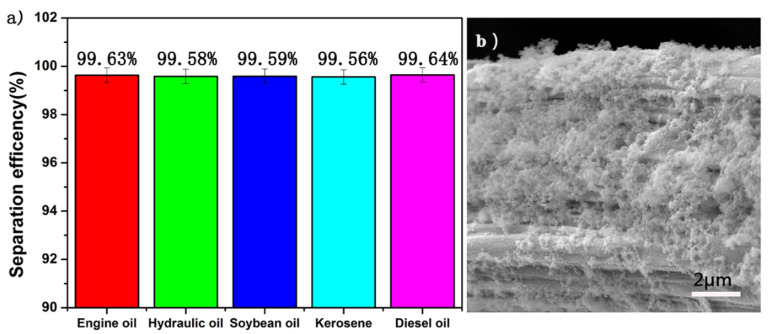
(**a**) The separation efficiency of SiO_2_-Na_2_SiO_3_-coated SSF after sandpaper abrasion and (**b**) the corresponding SEM image.

**Figure 7 materials-14-02628-f007:**
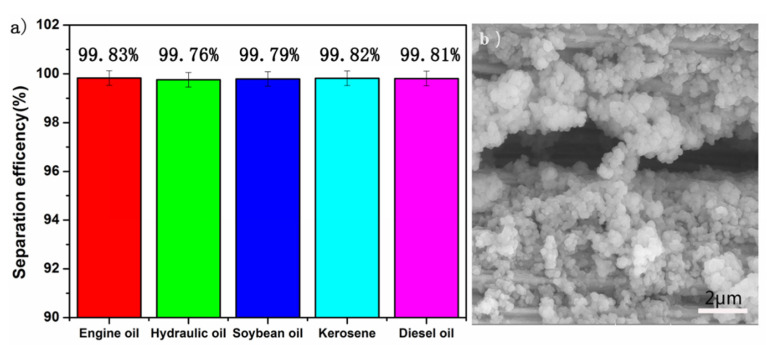
(**a**) The separation efficiency of SiO_2_-Na_2_SiO_3_-coated SSF after being immersed in hot water for 2 weeks and (**b**) corresponding SEM image.

**Figure 8 materials-14-02628-f008:**
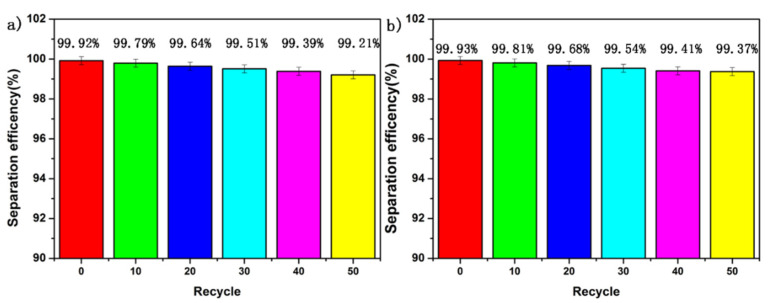
Variation of separation efficiency with times under salt (**a**) and acid (**b**) environments.

**Figure 9 materials-14-02628-f009:**
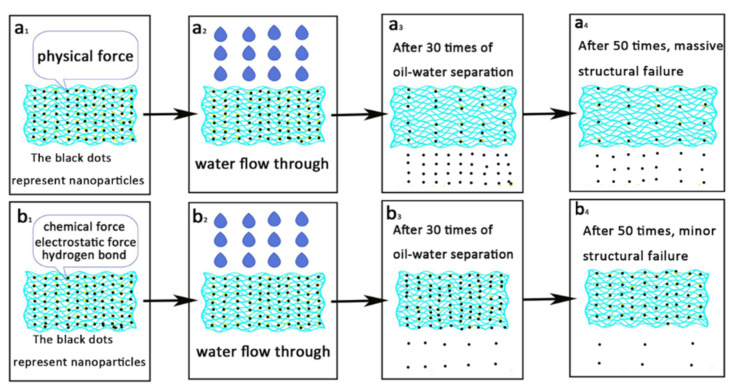
(**a_1–4_**) illustration of the ordinary oil–water membrane separation process, and (**b_1–4_**) illustrate separation using the superhydrophilic-underwater superoleophobic membrane as reported in this article.

## Data Availability

The data presented in this study are available on request from the corresponding author.

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
