# Peer review of "Synthesis of Si-Based High-Efficiency and High-Durability Superhydrophilic-Underwater Superoleophobic Membrane of Oil–Water Separation"

_materials, 2021, doi:10.3390/ma14102628_

Round 1
Reviewer 1 Report
An interesting article:
Some points must be adressed:
Change:
We suggest that the improved durability may be due the presence of sodium silicate that enhanced the membrane strength
with
We suggest that the improved durability may be due to the presence of sodium silicate that enhanced the membrane strength
I do not understand this phrase:
and then grafted the surface with a hydrophilic 2-mercaptoethyl containing carboxyl group (containing carboxyl group) to make a CF membrane for oil water separation.
Change
Wang et al. grafted Kaolin particles onto a stainless mesh
With
Wang et al. grafted Kaolin particles onto a stainless-steel mesh
Explain what means PVA, VPOSS, ASP, PDA, PVDF-HFP, CA, PEI , KH_560, etc. Explain all abbreviations!
Change:
2.1. Materials with 2.1.Materials and synthesis
Change:
The stainless steel fiber was cut into 4 cm × 4 cm pieces
With
The stainless-steel mesh was cut into 4 cm × 4 cm pieces
Please correct:
After 40 mins, SSF were taken out, washed with what and dried in an oven at 85 ℃ for 24 h
Please correct the figure captions of Figure 1: (b) instead of (a)
Change EDS with EDX
Change:
Therefore, that the SSF surface changed from hydrophobic to superhydrophilic after the surface treatment and had underwater superoleophobic properties.
With
Therefore, the SSF surface changed from hydrophobic to superhydrophilic after the surface treatment and had underwater superoleophobic properties.
Change:
due to underwater super oleophobicity of the membrane makes the oil impenetrable,
With:
due to the fact that underwater super oleophobicity of the membrane makes the oil impenetrable,
Change:
As shown in Fig. 4(b), after 50 recycles,
With:
As shown in Fig. 4(b), after 50 cycles,
Change:
showing good oil-water separation performance, as shown in Fig. 4(c), the SiO2 nanoparticles still present on the surface.
With
showing good oil-water separation performance and, as shown in Fig. 4(c), the SiO2 nanoparticles are still present on the surface.
Change:
We use the above formula (2) to calculate the water flux.
With
The formula (2) was used to calculate the water flux.
Change:
As shown in Fig. 5(b). The pristine SSF
With
As shown in Fig. 5(b), the pristine SSF
Change:
immersed in hot water for 2 week
with
immersed in hot water for 2 weeks
Corect in conclusions:
Tetrachloeomethane
With
tetrachloromethane
Author Response
Dear Professor/editor,
Thank you very much for your attention to our article “Synthesis of Si-based high-efficiency and high-durability superhydrophilic-underwater superoleophobic membrane of oil-water separation” (Manuscript ID:materials-1186678 ). The comments from referees are very valuable and helpful for improving our article. All the authors have seriously discussed about these comments. We have accordingly revised the manuscript to meet with the requirements of the journal “materials”. Our changes are included in the revised manuscript marked by the RED font and the detailed point-by-point responses to the comments are listed as RED font as following.
----------------------------------------------------------------------------------------
To reviewer: 1
We have revised and marked the sentence problem you raised, please review it again, if there is still a problem, please let me know.
Thank you for your positive comments and valuable suggestions to improve the quality of our manuscript.
----------------------------------------------------------------------------------------
Again, many thanks for your editorial endeavors on behalf of all of the authors and we do appreciate very much the constructive comments and good suggestions from the reviewers such that we were able to improve the overall quality and clarity of our paper. Hopefully, we could have our article been considered of publication in your journal. We really believe this work is sufficient novelty and impact to appeal to your readership. Should there been any other corrections we could make, please feel free to contact us.
Sincerely Yours,
Xiao-hui Fang
College of Materials and Chemistry Engineering,
Hainan University,
Haikou 570228, PR China.
E-mail: fangxiaohui110@sina.com
Reviewer 2 Report
Fang and co-workers present a new approach to produce membranes for oil-water separation. Their approach is based on coating stainless steel fibers (SSFs) with silicon dioxide to change the wettability of SSFs from hydrophobic to hydrophilic and oleophobic. Even though excellent performance is reported for separating different kinds of oils from water, further discussion and interpretation are required to make the manuscript suitable for publication in Materials, and thus, it cannot be published in its current form. Below you find the comments:
- Separation efficiencies reported in Figures 4, 6-8 are all larger than 99%. While this is a great performance, the definition in Eq. 1 cannot support the reported efficiencies of >99% unless the amount of water in the feed is tiny—not supported by Figure 4d. There are two possible approaches to address this question: i. m_0 in Eq. 1 is not the total mass of water and oil, and instead, it is the initial mass of oil and ii. The definition in Eq. 1 is correct and the values reported in the figures need to be recalculated. Also, the provided reference [36] does not report Eq. 1. Please justify.
- In the surface wettability tests shown in Figure 3, the authors used inverted contact angle measurements for n-hexane, paraffin oil, and hexadecane as they have smaller density than water. How did the authors conduct the sliding angle for these light oils shown in Figure 3b?
- In lines 216-220, the authors discussed the self-cleaning properties of coated SSF. The petri dishes in Figure 5b look wet. What is the liquid beneath the SSF? Can they elaborate the mechanism behind the self-cleaning process?
- In lines 227-230, the authors discussed the reduction of SiO2 nanoparticles density. Can they explain how did they determine this conclusion and can they quantify the reduction?
- In lines 254-256, the authors described how the membrane failed under basic conditions by detecting a color change. Can the author add after how many cycles the damage happened?
- The intention in Figure 9 is to compare conventional membrane to the reported one. While the first two panels for each case, a1/b1 and a2/b2, are identical, the differences in panels a3/b3 are not clear. What are the yellow dots? What does happen to oil? How is the failure demonstrated in a3/b3? A different schematic for Figure 9 is highly recommended.
- Scale bars in SEM images in Figures 1, 4, 6, and 7 are not shown.
- Is “Degree” in Figure 2b 2theta? If so, please revise.
- Line 153, “Figs 1c and 1” needs to revise.
Author Response
Dear Professor/editor,
Thank you very much for your attention to our article “Synthesis of Si-based high-efficiency and high-durability superhydrophilic-underwater superoleophobic membrane of oil-water separation” (Manuscript ID:materials-1186678 ). The comments from referees are very valuable and helpful for improving our article. All the authors have seriously discussed about these comments. We have accordingly revised the manuscript to meet with the requirements of the journal “materials”. Our changes are included in the revised manuscript marked by the RED font and the detailed point-by-point responses to the comments are listed as RED font as following.
---------------------------------------------------------------------------------------
To reviewer: 2
Comments:
- Separation efficiencies reported in Figures 4, 6-8 are all larger than 99%. While this is a great performance, the definition in Eq. 1 cannot support the reported efficiencies of >99% unless the amount of water in the feed is tiny—not supported by Figure 4d. There are two possible approaches to address this question: i. m_0 in Eq. 1 is not the total mass of water and oil, and instead, it is the initial mass of oil and ii. The definition in Eq. 1 is correct and the values reported in the figures need to be recalculated. Also, the provided reference [36] does not report Eq. 1. Please justify.
The expression is not accurate, m0 represents the mass of water in the oil-water mixture, and m1 represents the mass of water collected after separation. The reason that the document [36] has no formula is that an error occurred when the reference was added, and it has been adjusted.
In the surface wettability tests shown in Figure 3, the authors used inverted contact angle measurements for n-hexane, paraffin oil, and hexadecane as they have smaller density than water. How did the authors conduct the sliding angle for these light oils shown in Figure 3b?
When testing the sliding angle, the oil we selected is not exactly the same as the oil when measuring the contact angle. The negligence in the text is not expressed. When drawing the drawing, we forgot to change the abscissa, and the abscissa in Figure 3(a) was used. An error occurred. It has been explained and modified in the text.
In lines 216-220, the authors discussed the self-cleaning properties of coated SSF. The petri dishes in Figure 5b look wet. What is the liquid beneath the SSF? Can they elaborate the mechanism behind the self-cleaning process?
There is water in the petri dish. The sample is soaked with water first. The oil droplets can adhere to the original stainless steel fiber literature, but not to the SiO2-Na2SiO3-coated SSF. We refer to the literature [41], from [41] Find out the principle of self-cleaning.
4.In lines 227-230, the authors discussed the reduction of SiO2 nanoparticles density. Can they explain how did they determine this conclusion and can they quantify the reduction?
We can see from the comparison of Fig. 1(d) and Fig. 6(b) that before the friction and wear test, the SiO2 nanoparticles are completely coated on the surface of the stainless steel fiber, and after the friction and wear , A lot of SiO2 nano-particles fall off, resulting in a decrease in the number of SiO2 nano-particles that can be observed, and a large amount of the surface of the base stainless steel fiber.
5.In lines 254-256, the authors described how the membrane failed under basic conditions by detecting a color change. Can the author add after how many cycles the damage happened?
In the first oil-water separation, the separation efficiency is about 70%, and then the separation cannot be carried out, so only adding a separation cycle does not make much sense
6-8 Comments
Modifications have been made in the text, if there are still problems, please let me know
9.Line 153, “Figs 1c and 1” needs to revise.
I did not find "Figs 1c and 1" in line 153. If there are still problems, please continue to contact me
Thanks for your nice suggestions. We are really sorry for the careless mistakes. This manuscript has been double-checked by a native English speaker. In the marked revised version, some grammatical and spelling changes to our manuscript were all highlighted by using red colored text. We believe that the manuscript has greatly benefit from this revising.
----------------------------------------------------------------------------------------
Again, many thanks for your editorial endeavors on behalf of all of the authors and we do appreciate very much the constructive comments and good suggestions from the reviewers such that we were able to improve the overall quality and clarity of our paper. Hopefully, we could have our article been considered of publication in your journal. We really believe this work is sufficient novelty and impact to appeal to your readership. Should there been any other corrections we could make, please feel free to contact us.
Sincerely Yours,
Xiao-hui Fang
College of Materials and Chemistry Engineering,
Hainan University,
Haikou 570228, PR China.
E-mail: fangxiaohui110@sina.com
Reviewer 3 Report
The Authors reported the fabrication of SiO2-Na2SiO3-coated stainless steel fibers (SSF) via a one-step hydrothermal process. The material exhibits and high water/oil separation efficiency.
The topic of the manuscript is within the scope of the journal. It is rather well written and conclusions are supported by the results.
Therefore, I can recommend it for publication after adressing the following issues.
- please specify the stainless steel starting material
- it is recommended to remove/change a deep blue background in figure 2a since the figure is not informative enough in the present form
- are the CA values averaged? If so, from how many measurements? what is the reproducibility of the results. Error bars should be added to Figure 3a, and b
- from how many measurements was the oil/water separation efficiency determined? An error bars should be added to Fig. 4a, b, Fig. 5a, Fig. 6a, Fig. 7a, and Fig. 8
- a clearly visible scale bars should be added to SEM images
- a language/technical revision of the manuscript is also recommended
Author Response
Dear Professor/editor,
Thank you very much for your attention to our article “Synthesis of Si-based high-efficiency and high-durability superhydrophilic-underwater superoleophobic membrane of oil-water separation” (Manuscript ID:materials-1186678 ). The comments from referees are very valuable and helpful for improving our article. All the authors have seriously discussed about these comments. We have accordingly revised the manuscript to meet with the requirements of the journal “materials”. Our changes are included in the revised manuscript marked by the RED font and the detailed point-by-point responses to the comments are listed as RED font as following.
----------------------------------------------------------------------------------------
To reviewer: 3
Comments:
- please specify the stainless steel starting material
- it is recommended to remove/change a deep blue background in figure 2a since the figure is not informative enough in the present form
- are the CA values averaged? If so, from how many measurements? what is the reproducibility of the results. Error bars should be added to Figure 3a, and b
- from how many measurements was the oil/water separation efficiency determined? An error bars should be added to Fig. 4a, b, Fig. 5a, Fig. 6a, Fig. 7a, and Fig. 8
- a clearly visible scale bars should be added to SEM images
- a language/technical revision of the manuscript is also recommended
Both CA and separation efficiency are the average values after 5 measurements. Regarding the problems in Figures 2a, 3a-b, 4a, 5a, 6a, 7a and 8 raised by the reviewer, the text has been revised. If there are still problems, please tell me
We are really sorry for the careless mistakes. Thank you for your nice reminding. We have carefully corrected these errors in revised manuscript.
----------------------------------------------------------------------------------------
Again, many thanks for your editorial endeavors on behalf of all of the authors and we do appreciate very much the constructive comments and good suggestions from the reviewers such that we were able to improve the overall quality and clarity of our paper. Hopefully, we could have our article been considered of publication in your journal. We really believe this work is sufficient novelty and impact to appeal to your readership. Should there been any other corrections we could make, please feel free to contact us.
Sincerely Yours,
Xiao-hui Fang
College of Materials and Chemistry Engineering,
Hainan University,
Haikou 570228, PR China.
E-mail: fangxiaohui110@sina.com
Round 2
Reviewer 2 Report
Thanks for addressing the comments! The manuscript has been improved and it is ready for publication. Here are a few minor comments:
- “Fig. 1c and 1.” is now in line 159.
- Line 111, “SSF was taken out…”
- Line 129, extra comma after “separation”.
I recommend another round of proofreading to fix any grammar, syntax, spelling, and punctuation errors.